# The Effects of Alternative Foods on Life History and Cannibalism of *Amblyseius herbicolus* (Acari: Phytoseiidae)

**DOI:** 10.3390/insects13111036

**Published:** 2022-11-09

**Authors:** Fei Hou, Zhao-Hong Ni, Meng-Ting Zou, Rui Zhu, Tian-Ci Yi, Jian-Jun Guo, Dao-Chao Jin

**Affiliations:** Institute of Entomology, Guizhou University, Scientific Observing and Experimental Station of Crop Pest in Guiyang, Ministry of Agriculture and Rural Affairs of the P. R. China, Guiyang 550025, China

**Keywords:** biological control, mass rearing, pollen, food quality

## Abstract

**Simple Summary:**

Suitable foods are essential for the successful mass rearing of natural enemies and can directly affect their quality and performance, which are the determinants of the success of biological control programs. Most predatory mites of the family *Phytoseiidae* are omnivorous, meaning they can obtain nutrients from both prey and plant sources. However, different kinds or species of alternative foods have significantly different effects on the development, survivorship and reproduction of predatory mites. Meanwhile, food quality also significantly affects the cannibalistic behavior of predatory mites, which limits the application of alternative foods in the mass rearing of the predators. *Amblyseius herbicolus* (Chant), an excellent predator of various phytophagous mites and insects, has recently gained attention as a crucial biocontrol agent. We evaluated the effects of *A. herbicolus* feeding on six alternative foods (*Oulenziella bakeri*, *Tyrophagus putrescentiae*, *Aleuroglyphus ovatus*, almond pollen (*Prunus armeniaca*), apple pollen (*Malus pumila*), maize pollen (*Zea mays*)), and natural prey (*Tetranychus urticae*) on its life history and cannibalism. Our findings indicated that *O. bakeri* and almond pollen performed best for *A. herbicolus* as potential foods for mass rearing and for supporting its population after release in the fields.

**Abstract:**

The development, survivorship, fecundity, and cannibalism of the predatory phytoseiid mite, *Amblyseius herbicolus* (Chant), fed six different alternative foods (*Oulenziella bakeri*, *Tyrophagus putrescentiae*, *Aleuroglyphus ovatus*, almond pollen (*Prunus armeniaca*), apple pollen (*Malus pumila*), maize pollen (*Zea mays*)), and natural prey (*Tetranychus urticae*) were determined in the laboratory. Our findings indicated that *A. herbicolus* that fed on all six alternative foods could normally complete its developmental and reproductive cycles. The shortest pre-adult developmental duration was observed when *A. herbicolus* fed on almond pollen (4.91 d) as well as *T. urticae* (4.90 d), and the longest when it fed on maize pollen (6.24 d). Pre-adult survival rates were higher when the predator fed on almond pollen (0.99), maize pollen (0.96), and *O. bakeri* (0.93). The highest fecundity was observed when *A. herbicolus* fed on apple pollen (28.55 eggs/female), almond pollen (26.06 eggs/female), and *O. bakeri* (26.02 eggs/female) in addition to *T. urticae* (48.95 eggs/female), and the lowest when it fed on maize pollen (7.84 eggs/female). The highest value of the intrinsic rate of increase (*r*) was obtained when *A. herbicolus* fed on *O. bakeri* (0.202 d^−1^) in addition to *T. urticae* (0.210 d^−1^), followed by almond pollen (0.163 d^−1^), and the lowest was when it fed on maize pollen (0.064 d^−1^). Cannibalism of conspecific eggs by adults of *A. herbicolus* did not occur when *O. bakeri* and *T. urticae* were provided. The cannibalism rate of the predatory mite was the lowest when fed on almond pollen, *T. putrescentiae,* and *A. ovatus* and the highest on apple pollen. Above all, when fed on *O. bakeri* and almond pollen, and with no or low cannibalism rate, *A. herbicolus* had the best development, survivorship, fecundity, and population parameters. Therefore, *O. bakeri* and almond pollen could be potential alternative foods for mass rearing programs of *A. herbicolus* or to support its population in the fields.

## 1. Introduction

Natural enemies are an important component in integrated pest management programs [1,2]. However, the application of natural enemies in biological control is often limited. One of the main issues is that natural enemies are difficult to mass-rear [3], and another obstacle is the population establishment and persistence of released natural enemies in the fields [4,5]. Alternative food for natural enemies has become an important strategy in biological control programs for its low cost for natural enemy mass production [6,7,8]. Meanwhile, the survival or populations of natural enemies can be maintained by providing alternative foods in the fields when the target prey is scarce [2,9], and the performance of biological control can be improved [7,8]. Hence, alternative foods play a crucial role in the application of natural enemies.

The family Phytoseiidae includes many species of generalist predators that can develop and reproduce when various foods are provided as alternatives to their natural prey [6,10,11,12]. Nevertheless, alternative foods do not always have a positive effect, and poor-quality alternative foods may result in slow growth and development [8,9,13,14], as well as the decreased survival and fecundity of predatory mites [15,16,17]. Conversely, after the long-term mass rearing of omnivorous predatory mites with poor-quality alternative foods, cannibalism will occur, significantly affecting the structure and population dynamics of the predator [18,19,20]. High-quality alternative foods can significantly reduce the cannibalism of predatory mites [6,10,21]. Therefore, suitable and high-quality alternative foods are the prerequisite for the successful mass rearing of omnivorous predatory mites for biological control. Currently, non-pest foods, such as pollen, artificial diets, or relatively harmless arthropods, are mainly considered alternative foods for omnivorous predatory mites [11,22,23,24].

*Amblyseius herbicolus* (Chant) (Acari: Phytoseiidae) is an omnivorous predatory mite [25] that has been found in many agricultural ecosystems, including citrus, coffee, and pepper [26,27,28], to predate several phytophagous pest mites and insects, such as *Oligonychus ilicis* (McGregor) (Acari: Tetranychidae) [29], *Brevipalpus phoenicis* (Geijskes) (Acari: Tenuipalpidae) [30], *Polyphagotarsonemus latus* (Banks) (Acari: Tarsonemidae) [27], *Diaphorina citri* Kuwayama (Hemiptera: Psyllidae) [25], *Sericothrips staphylinus* Haliday (Thysanoptera: Thripidae) [31,32], and *Bemisia tabaci* (Gennadius) (Hemiptera: Aleyrodidae) [33]. It is currently regarded as an important biocontrol agent [34].

However, *A. herbicolus* was limited in application because the known alternative foods were less suitable than the preferred natural prey. Hence, we investigated three prey mites (*Oulenziella bakeri* (Hughes) (Acari: Winterschmidtiidae), *Tyrophagus putrescentiae* (Schrank) (Acari: Acaridae), and *Aleuroglyphus ovatus* (Troupeau) (Acari: Acaridae)) and three plant pollens (*Prunus armeniaca* L. (Rosales: Rosaceae), *Malus pumila* Mill. (Rosales: Rosaceae), and *Zea mays* L. (Poaceae)) as alternative foods, and *Tetranychus urticae* Koch (Acari: Tetranychidae) as a natural prey served as the control group to assess the effect of alternative foods on the life history and cannibalism of *A. herbicolus*. The results will provide information that will improve our knowledge of its mass rearing and food supplementation in the fields.

## 2. Materials and Methods

### 2.1. Biological Materials

Predatory mite: *A. herbicolus* was collected from an *Osmanthus* sp. tree on the South Campus of Guizhou University (Huaxi, Guiyang, China, 26°42′94″ N, 106°66′78″ E) and reared in 2–3 generations in a mite-rearing cell in an artificial climate box with *O. bakeri* as the prey under conditions of 25 ± 2 °C, 85 ± 5% relative humidity (RH), and 16:8 h light/dark photoperiod.

Alternative foods: *O. bakeri* and *A. ovatus* were from Fuzhou Guannong Biological Science and Technology Co. Ltd., Fuzhou, China; *T. putrescentiae* was obtained from abandoned flour from a flour miller at Huaxi, Guiyang, China; all mites were maintained on yeast powder (Angel Yeast Co. Ltd., Yichang, China) in a mite-rearing cell in an artificial climate box under conditions of 25 ± 2 °C, 85 ± 5% RH, and 16:8 h light/dark photoperiod. Apple and almond pollen were purchased from Qingdao Jinbaolun Agricultural Science and Technology Co., Ltd., Qingdao, China; maize pollen was collected from a corn field in Huaxi, Guiyang, China, and dried in an oven (35 °C) for 48 h. All plant pollens were stored at −20 °C in a freezer.

Natural prey (the control group): *T. urticae* is a 12-year experimental population raised on common bean *Phaseolus vulgaris* in an artificial climate chamber under conditions of 25 ± 5 °C, 60 ± 10% RH, and 14:10 h light/dark photoperiod.

### 2.2. Experimental Apparatus

Mite-rearing cell: The rearing cell of *A. herbicolus* was composed of two layers of plexiglass slide, the bottom layer (6 cm × 5 cm × 0.5 cm) with a circular hole in the middle (4-cm diameter); a 200-mesh black gauze was glued to the bottom of the hole, and a suitable amount of sewing thread was placed as the egg-laying substrate for predatory mites. The cell was covered on the top layer with plexiglass slide (6 cm × 5 cm × 0.5 cm), and both ends of the cell were clamped with dovetail clamps.

Mite-rearing box: The mite-rearing box is a plastic box (18 cm × 18 cm × 8 cm). In the box, there was a sponge platform (9-cm diameter, 2.0-cm height) on which a dish (9-cm diameter) was placed with initial population of alternative preys. The distilled water was poured into the box to the height of the sponge as a barrier to prevent prey mites from escaping. Then, a hole (9 cm × 9 cm) on the lid of the box was punched out and the hole was covered with a piece of 200-mesh gauze bigger than the hole by glue.

Experimental unit: When *T. urticae* served as the prey, the unit consists of three layers of plexiglass slides. Both the middle of the top layer and the middle layer (4 cm × 3 cm × 0.3 cm) had a hole (2-cm diameter), and only the hole in the top layer was covered with a 200-mesh black gauze. The bean leaves (about 4 cm × 3 cm) with *T. urticae* were placed between the bottom (4 cm × 3 cm × 0.2 cm) and middle layers, then the top layer was covered, and both ends were clamped with dovetail clips. The experimental unit for other foods was composed of two layers of plexiglass slides, with the same specifications as the top and bottom layers of the unit when *T. urticae* served as the prey.

### 2.3. Effects of Different Alternative Foods on Life Table Parameters

To obtain same-age eggs of *A. herbicolus*, 60 mated *A. herbicolus* adults were transferred from the colonies into a new mite-rearing cell with enough *O. bakeri* as food. After 24 h, each newly laid egg was separated and collected into other new experimental units.

Mixed life stages of each prey mite (*T. urticae*, *O. bakeri*, *A. ovatus*, and *T. putrescentiae*): pollen (apple, almond, and maize), and a same-age egg were added to each experimental unit to assess the effect of alternative foods on the life table parameters of *A. herbicolus*. Developmental stage and survival rate of the predatory mite were observed and recorded twice daily (8:00 and 20:00) until the adult stage.

Emerged male and female adults of *A. herbicolus* were paired, and each pair was transferred to a new experimental unit with the corresponding foods; about 1.0 cm of sewing thread was provided for females to lay eggs. The same-age males from the rearing cell of the corresponding foods were used to mate with females from the life table cohort if there were more females than males, but males in the rearing cell were omitted from the life table analysis. Daily oviposited eggs of each unit were recorded and discarded from the unit. The survival of each adult was recorded daily until the death of all individuals. A life table study was conducted of *A. herbicolus* that fed on different foods under conditions of constant temperature 25 ± 2 °C, 85 ± 5% RH, and 16:8 h light/dark photoperiod.

### 2.4. Effects of Different Alternative Foods on Cannibalism

To evaluate the effects of different alternative foods on the cannibalism of *A. herbicolus*, eggs of this predator were used as the stage to be cannibalized by adult females. A three-day-old *A. herbivorous* female was introduced into a new experimental unit with ten conspecific eggs (<24 h) reared on *O. bakeri* and corresponding alternative foods (15 mg of each pollen, 60–80 individuals of prey mites) and natural prey (60–80 individuals of *T. urticae* on 15-mm-diameter bean leaves).

The number of cannibalized eggs was recorded daily for seven days using a stereoscopic microscope (3.35–300×, Nikon SMZ745, Shanghai, China). Daily, the predator mite needed to be moved to a new experimental unit with ten new conspecific eggs and corresponding food. Cannibalized eggs were recognized by the shell from which the internal egg content was removed [6]. Fifteen replicates were performed for each diet. The experimental unit was placed in an artificial climate box under conditions of 25 ± 2 °C, 85 ± 5% RH, and 16:8 h light/dark photoperiod.

### 2.5. Statistical Analyses

Life table analysis: One-way analysis of variance (ANOVA) was used to analyze developmental durations, longevity, and fecundity of *A. herbicolus* that fed on different diets, and Tukey’s honestly significant difference was used to compare the differences among different foods (SPSS Statistics v.23.0, IBM). The variances and standard error of the survival rates and population parameters were estimated using the bootstrap procedure with 100,000 resamplings to obtain stable estimates [35,36]. Comparison of the survival rates and population parameters was performed using the paired bootstrap test in the TWOSEX-MS Chart program (National Chung Hsing University) [37]. The following parameters were estimated: the age–stage-specific survival rate (*S_xj_*) (where *x* is age, *j* is stage); the age–stage-specific fecundity (*f_xj_*); the age-specific survival rate (*l_x_*); the age-specific fecundity (*m_x_*), and the population growth parameters (the intrinsic rate of increase (*r*); the finite rate of increase (*λ*); the net reproductive rate (*R*_0_), and the mean generation time (*T*)).

The intrinsic rate of increase (*r*) was estimated using iterative bisection and the Euler–Lotka equation, with the age indexed from 0:$$\overset{\infty }{\underset{x=0}{\sum }}e^{-r\left(x+1\right)}l_{x}m_{x}$$

In the age–stage, two-sex life table, the *l_x,_* and *m_x_* are calculated as:$$l_{x}=\sum_{j=1}^{m}S_{xj}\;l_{x}=\sum_{j=1}^{m}S_{xj}$$
where *m* is the number of stages.

The finite rate of increase (*λ*), the mean generation time (*T*), and the net reproductive rate (*R_0_*) were calculated as:$$\lambda =e^{r}\;T=\ln R_{0}/r\;R_{0}=\sum_{x=0}^{\infty }l_{x}m_{x}$$

Cannibalism data analysis: The Kolmogorov–Smirnov test revealed that the data on the effect of diet treatments and predation days on cannibalism were not normally distributed; therefore, cannibalism was evaluated with a nonparametric two-factor ANOVA using R (v.4.1.1, R Core Team). The effect of diet types with seven levels and predation days with seven levels were then analyzed using the function “scheirerRayHare” package in R. We used the nonparametric Kruskal–Wallis H test to determine the mean and significance of the different treatments for the factors that had a significant effect on cannibalism data (SPSS Statistics v.23.0, IBM). All mean comparisons with *p* < 0.05 were considered significantly different.

## 3. Results

### 3.1. Developmental Durations and Reproduction

*A. herbicolus* that fed on different alternative foods showed different developmental durations (Table 1). There were significant differences in pre-adult duration (*F*_6,547_ = 102.737, *p* < 0.0001) on different foods: The shortest pre-adult duration was recorded when *A. herbicolus* fed on almond pollen (4.91 d), in addition to *T. urticae* (4.90 d), and the longest was on maize pollen (6.24 d). However, there were no significant differences between *O. bakeri*, *T. putrescentiae,* and *A. ovatus*: 5.41, 5.32, and 5.43 d, respectively. More specifically, significant differences were recorded in larval (*F*_6,580_ = 137.312, *p* < 0.0001), protonymph (*F*_6,558_ = 53.364, *p* < 0.0001), and deutonymph (*F*_6,547_ = 39.311, *p* < 0.0001) durations on different foods. The shortest larval duration was recorded when *A. herbicolus* fed on almond pollen, and it showed no significant difference from *O. bakeri* and *T. urticae*, whereas the longest was obtained on maize pollen. The shortest protonymph duration was observed when *A. herbicolus* fed on almond pollen, and the longest duration was recorded when it fed on maize pollen, while protonymph duration was not significantly different between *O. bakeri* and *T. putrescentiae* and was shorter than *A. ovatus*. The shortest deutonymph duration was observed when *A. herbicolus* fed on maize pollen, and the longest duration was on *O. bakeri*. There were no significant differences in egg development duration (*F*_6,644_ = 0.915, *p* = 0.4580) on different foods because the predator eggs tested came from a population of *A. herbicolus* reared on the same prey. Meanwhile, the total lifespan of the predator differed among the six tested alternative foods and ranged from 24.74 (*T. putrescentiae*) to 44.72 d (apple pollen) (*F*_6,547_ = 58.930, *p* < 0.0001). Adult longevities (female, *F*_6,288_ = 67.027, *p* < 0.0001; male, *F*_6,252_ = 73.752, *p* < 0.0001) of *A. herbicolus* fed on natural prey and plant pollen were significantly longer than when it fed on prey mites; the longest female and male adult longevities were recorded when *A. herbicolus* fed on apple pollen and maize pollen, respectively; the shortest female adult longevity was recorded when *A. herbicolus* fed on *O. bakeri*, whereas the shortest male adult longevity was when it fed on *T. putrescentiae* and *A. ovatus*.

Furthermore, the shortest duration of adult preoviposition period (APOP) (*F*_6,283_ = 30.709, *p* < 0.0001) and total preoviposition period (TPOP) (*F*_6, 283_ = 51.084, *p* < 0.0001) were recorded when *A. herbicolus* fed on *T. urticae*. The longest ovipositional period (*F*_6, 283_ = 213.939, *p* < 0.0001) was observed when *A. herbicolus* fed on apple and almond pollen in addition to *T. urticae*, followed by *O. bakeri*, and the shortest was when it fed on maize pollen, *T. putrescentiae,* and *A. ovatus*. The highest fecundity (*F*_6, 283_ = 358.226, *p* < 0.0001) was recorded when *A. herbicolus* fed on apple pollen (28.55 eggs/female), almond pollen (26.06 eggs/female), and *O. bakeri* (26.02 eggs/female) in addition to *T. urticae* (48.95 eggs/female), and no differences were observed among the values obtained on the three alternative foods, while the lowest was observed on maize pollen (7.84 eggs/female).

### 3.2. Survival Rate and Fecundity Curves

The age–stage-specific survival rate (*S_xj_*) curves for the pre-adult and adult stages of *A. herbicolus* fed on six alternative foods and natural prey *T. urticae* (Figure 1 and Figure 2) showed survival and stage differentiation as well as variable developmental rates. The overlap of stages during the developmental duration can also be observed in Figure 1 and Figure 2. The overlap among the different stages of the survival rates was not shown in the age-specific survivorship (*l_x_*) curves (Figure 3) compared with *S_xj_* curves (Figure 1 and Figure 2). The age-specific survivorship (*l_x_*) curves (Figure 3) of *A. herbicolus* significantly decreased twice when it fed on *T. urticae*, apple pollen, maize pollen, and *O. bakeri* and significantly decreased three times when it fed on *T. putrescentiae* and *A. ovatus*. Furthermore, pre-adult survival rates on almond pollen, maize pollen, and *O. bakeri* (0.99, 0.97, and 0.93, respectively) were higher than on apple pollen, *T. putrescentiae,* and *A. ovatus* (0.85, 0.76, and 0.62, respectively), while on its natural prey, *T. urticae*, it was 0.85 (*p* < 0.05) (Table 1). The highest survival rates of females (0.500) and males (0.490) were recorded when *A. herbicolus* fed on almond pollen, which were higher than that of maize pollen (same value of 0.482), *O. bakeri* (0.479 and 0.447, respectively), *T. urticae* (same value of 0.427), and apple pollen (same value of 0.423), and the values of female and male adults were the same or close. However, the survival values of female and male adults were quite different when *A. herbicolus* fed on *T. putrescentiae* (0.430 and 0.330, respectively) and *A. ovatus* (0.420 and 0.200, respectively), and male adults all died before 20 days in the predator mite population (Figure 2).

Figure 3 shows the age-specific fecundity (*m_x_*) and age–stage-specific fecundity (*f_xj_*) curves of *A. herbicolus* fed on six alternative foods and its natural prey, *T. urticae*. When *O. bakeri* was provided, the highest *f_xj_* and *m_x_* were 1.465 eggs female^−1^ day^−1^ and 0.753 eggs individual^−1^ day^−1^, respectively, which were higher than that of *T. urticae* (1.366 and 0.683, respectively). However, when maize pollen was provided, the highest *f_xj_* was 0.256 eggs female^−1^ day^−1^, and the highest *m_x_* was 0.128 eggs individual^−1^ day^−1^ of *A. herbicolus*, which were the lowest values of the predator fed on all the foods tested.

### 3.3. Population Parameters

The estimated life table parameters for *A. herbicolus* on six alternative foods and its natural prey, *T. urticae*, are presented in Table 2. The highest intrinsic rate of increase (*r*) was obtained when *A. herbicolus* fed on *O. bakeri* (0.202 d^−1^) in addition to *T. urticae* (0.210 d^−1^), followed by almond pollen (0.163 d^−1^), whereas the lowest was when it fed on maize pollen (0.064 d^−1^). The highest finite rate of increase (*λ*) was obtained when *A. herbicolus* fed on *O. bakeri* (1.224 d^−1^) in addition to *T. urticae* (1.233 d^−1^), and it was considerably lowest on maize pollen (1.066 d^−1^). The net reproductive rates (*R_0_*) of *A. herbicolus* fed on almond pollen, *O. bakeri,* and apple pollen were significantly higher than those on *A. ovatus*, *T. putrescentiae,* and maize pollen. The mean generation time (*T*) was longer when *A. herbicolus* fed on plant pollens than on prey mites.

### 3.4. Effects of Different Alternative Foods on Cannibalism

Nonparametric two-factor (diet types and predation days) ANOVA revealed that there was a significant effect of food type on cannibalism rate (Table 3). Cannibalism was observed when *A. herbicolus* fed on other foods in addition to *O. bakeri* and *T. urticae* (Figure 4). Specifically, the cannibalism rate on apple pollen was the highest (40%), followed by maize pollen (20%), and the lowest were on almond pollen, *T. putrescentiae,* and *A. ovatus* (10%).

## 4. Discussion

Polyphagous predatory mites can obtain nutrients from both prey and plant sources to support and maintain their populations and suppress pest populations [38]. It is well known that plant pollens and storage mites can be used as alternative foods for polyphagous predatory mites, but different kinds or species of alternative food have significantly different effects on the developmental, survival, and reproductive rates of predatory mites [22,38,39]. This study revealed that *A. herbicolus* fed on all six tested alternative foods could complete growth, development, and reproduction. Our findings indicated that *A. herbicolus* fed on almond pollen, apple pollen, and *O. bakeri* had positive outcomes, such as rapid development, high pre-adult survival rates, and fecundity. Almond and apple pollen also had similar positive effects on other predatory mites, such as *Typhlodromus foenilis* Oudemans [40], *T. Bagdasarjani* Wainstein and Arutunjan [41], *Neoseiulus cucumeris* (Oudemans) [42], and *A. Swirskii* (Athias-Henriot) [43]. In our experiments, when *A. herbicolus* fed on prey mite *O. bakeri*, its pre-adult duration was shorter than that when it was reared on the dried fruit mite, *Carpoglyphus lactis* (L.) according to Zhang and Zhang [34], but the fecundity was similar. Additionally, we observed that *A. herbicolus* fed on maize pollen had the shortest ovipositional periods and the lowest fecundity comparing the six tested alternative diets. These results were consistent with those of Ranabhat et al. [42], Delisle et al. [43], and Nguyen et al. [44], where maize pollen was less suitable for the reproduction of predatory mites. In our experiment and that of Delisle et al. [43], maize pollen appeared to absorb more ambient humidity than other pollen tested, so the accumulated pollen grains were not easily eaten by predatory mites, which may be one reason for the low fecundity and long pre-adult duration of *A. herbicolus* feeding on maize pollen. Additionally, our research showed low fecundity when *A. herbicolus* fed on the storage mites *T. putrescentiae* and *A. ovatus*. This result contradicted those of Asgari et al. [45] and Li et al. [46], as two predator mites *A. swirskii* and *N. barkeri* reared on storage mites *T. putrescentiae* and *A. ovatus*, respectively, both showed high fecundity. The lower fecundity of the feeding storage mites in our research may be attributable to factors [41] such as (1) certain nutrients insufficient to mite reproduction, (2) predator mite unable to acclimate to diet switching, and (3) predator mite inefficient to cope and feed on the larger storage mite individual.

The adult female and male longevities of *A. herbicolus* fed on plant resource foods were significantly longer than when it fed on alternative prey mites. However, the adult female longevity of *A. herbicolus* fed on apple pollen was not significantly different from when it fed on *T. urticae*, as well as adult male longevity of *A. herbicolus* fed on maize pollen and *T. urticae*. These results showed that plant pollen can prolong the adult longevity of predatory mites. Eini et al. [47] offered a list of plant pollen species that could increase the adult longevity of *N. californicus* in comparison to *T. urticae*. In our study, the adult female and male longevities of *A. herbicolus* that fed on *O. bakeri* were 20.38 and 19.12 days, respectively. Zhang and Zhang [34] reported similar results for the adult longevity of this predator fed on *C. lactis* (18–19 days). Our results also showed that the adult male longevities fed on *T. putrescentiae* and *A. ovatus* were 8.61 and 10.35 d, respectively, which is short compared with the adult female longevity. Additionally, the *S_xj_* curve showed that male adults in the predatory population that fed on *T. putrescentiae* and *A. ovatus* all died before 20 d. Some insects and mites have the habit of multiple mating to achieve maximum fecundity [48,49], and if male individuals are lost in the population too early, population growth cannot be achieved. This could also be a reason for the low fecundity of *A. herbicolus* fed the storage mites mentioned above; male individuals of the predatory mite died prematurely because they are difficult to copewith and feed on larger storage mites, causing female adults to fail to achieve maximum fecundity. Therefore, *T. putrescentiae* and *A. ovatus* are unsuitable as alternative preys for improving *A. herbicolus* populations.

The intrinsic rate of increase (*r*) is a key demographic parameter as it indicates the population growth potential [47]. Studies have shown that almond pollen is of high nutritional value for predatory mites [50,51,52,53]. Our results showed that the *r* of *A. herbicolus* fed on almond pollen (0.163 d^−1^) was higher than when it fed on other alternative foods tested in addition to *T. urticae* (0.210 d^−1^) and *O. bakeri* (0.202 d^−1^). Our value is similar to that of Riahi et al. [41] reported for *T. bagdasarjani* (0.1605 d^−1^). Nemati and Riahi [52] and Yazdanpanah et al. [53] reported, respectively, that *r* values of *A. swirskii* and *N. californicus* on almond pollen were even higher than the finding of the present study. Furthermore, the *r* and *λ* values of *A. herbicolus* fed on *T. urticae* and *O. bakeri* in our study are higher than those reported for this predator on castor bean pollen, sunnhemp pollen, broad mite *p. latus* [54] and tenuipalpid mite *B. phoenicis* [30].

Cannibalism of most predaceous natural enemies, such as predatory bugs [55,56], ladybugs [57], lacewings [58], and predatory mites [59] is inconducive to mass rearing programs for natural enemies [60]. Vangansbeke et al. [10] demonstrated that the cannibalistic behavior of omnivorous phytoseiid mites is diet-dependent. Marcossi et al. [6] also reported that high-quality alternative food reduces cannibalism in *A. herbicolus*. Likewise, the quality of different tested foods also significantly affected the cannibalistic behavior of *A. herbicolus* in our study. We demonstrated that cannibalism on conspecific eggs by adults of *A. herbicolus* did not occur when a suitable alternative prey (*O. bakeri*) and natural prey (*T. urticae*) were provided. Among other alternative foods, the lowest cannibalism rate of the predator was recorded when *A. herbicolus* fed on almond pollen, *T. putrescentiae, A. ovatus*, and maize pollen, while the highest was observed when it fed on apple pollen. Prey mite *O. bakeri* was reported as an important alternative prey for the commercial production of *N. californicus* [61], and combined with our results, *O. bakeri* could provide the necessary nutrients for *A. herbicolus*, thereby decreasing the probability of cannibalism during the mass rearing of *A. herbicolus*. Moreover, our results showed that *A. herbicolus* fed on almond pollen had only a lower cannibalism rate, whereas *A. herbicolus* fed on *O. bakeri* and almond pollen had a higher developmental rate, fecundity, and population parameters compared with other tested alternative foods. Therefore, *O. bakeri* and almond pollen can be considered potential alternative foods for the population growth of *A. herbicolus*.

## 5. Conclusions

Conclusively, our results demonstrated that *A. herbicolus* can use a broad range of foods, including natural prey spider mites, plant pollen, and alternative prey mites, and complete normal growth, development, and reproduction. Positive results, such as high developmental rate, fecundity, and population parameters, were observed when *A. herbicolus* fed on alternative prey *O. bakeri* and the plant resource almond pollen, which indicated the potential of both *O. bakeri* and almond pollen in mass rearing under controled conditions and maintaining populations in the field for *A. herbicolus*. However, the cannibalistic behavior on some tested foods and the potential impact on laboratory and field populations of the predator require further study.

## Figures and Tables

**Figure 1 insects-13-01036-f001:**
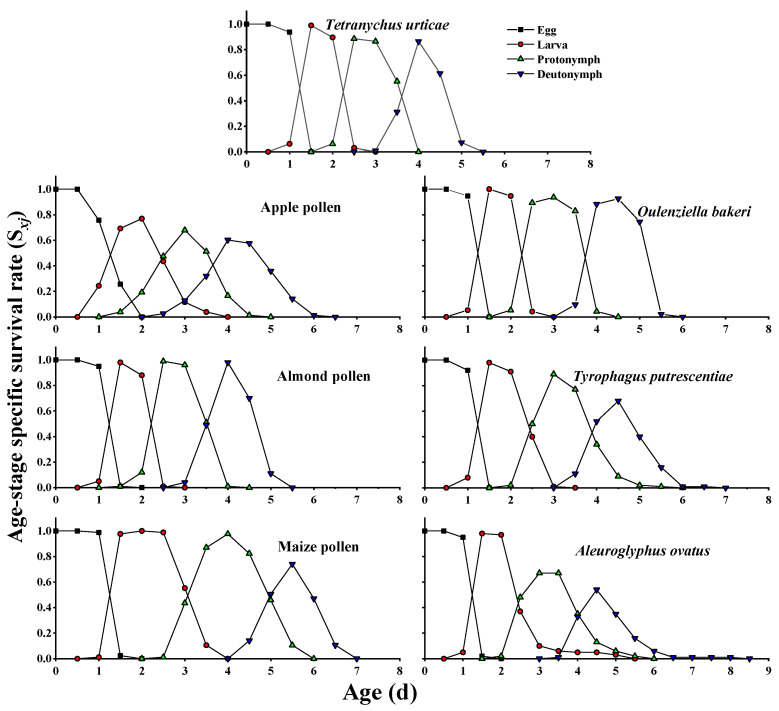
Age–stage-specific survival rate (*S_xj_*) of pre-adult stages of *A. herbicolus* on six alternative foods and its natural prey, *T. urticae*.

**Figure 2 insects-13-01036-f002:**
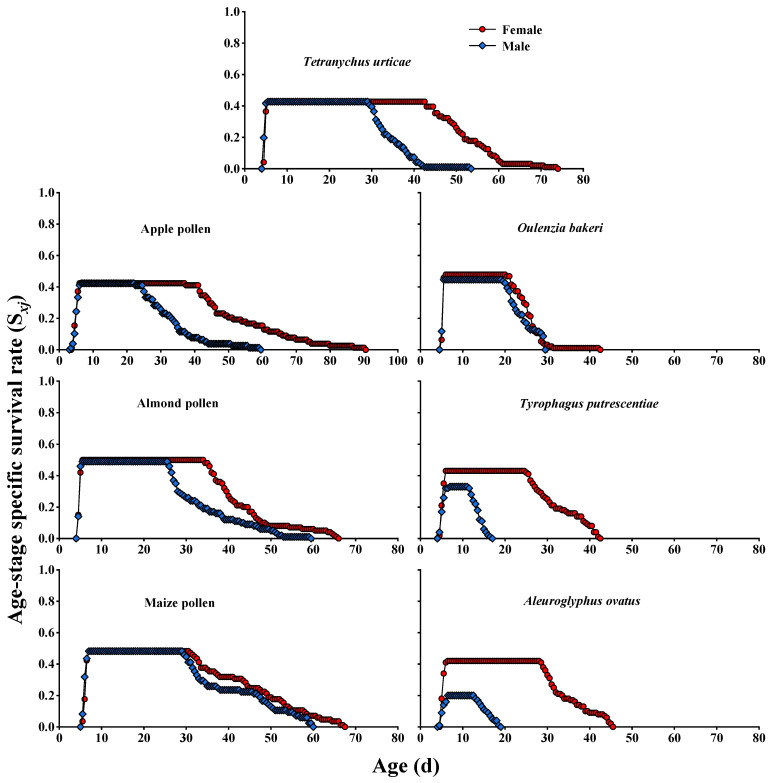
Age–stage-specific survival rate (*S_xj_*) of adult stages of *A. herbicolus* on six alternative foods and its natural prey, *T. urticae*.

**Figure 3 insects-13-01036-f003:**
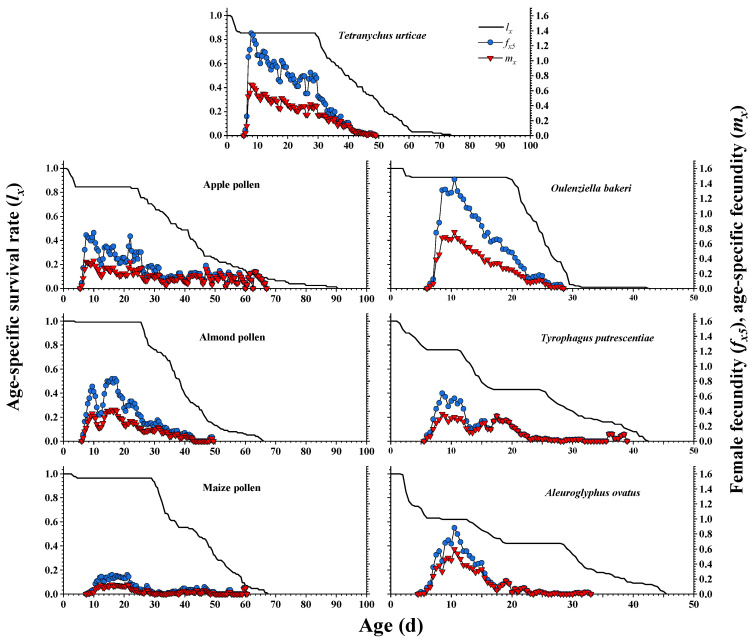
Age-specific survivorship (*l_x_*), age-specific fecundity (*m_x_*), and age–stage-specific fecundity (*f_xj_*) of *A. herbicolus* on six alternative foods and its natural prey, *T. urticae*.

**Figure 4 insects-13-01036-f004:**
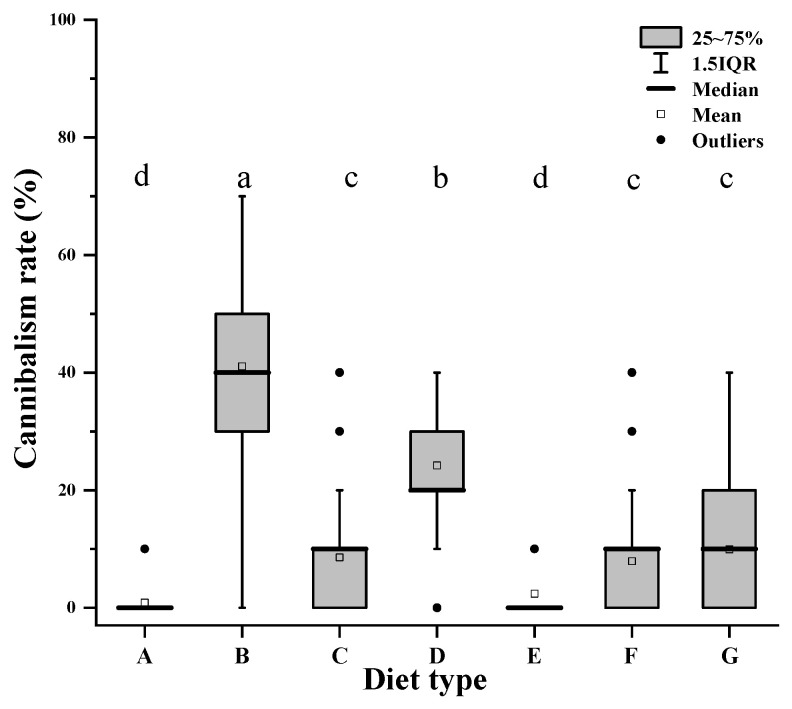
Effects of *T. urticae* (A), apple pollen (B), maize pollen (C), almond pollen (D), *O. bakeri* (E), *T. putrescentiae* (F), and *A. ovatus* (G) on the cannibalism of *A. herbicolus*. Different letters (a–d) showed the presence of significant differences following the Kruskal–Wallis H test (*p* < 0.05).

**Table 1 insects-13-01036-t001:** Development duration, longevity, and fecundity of *A. herbicolus* feeding on six alternative foods and its natural prey, *T. urticae*.

Parameters	Natural Prey	Alternative Food					
	*T. urticae*	Apple Pollen	Almond Pollen	Maize Pollen	*O. bakeri*	*T. putrescentiae*	*A. ovatus*
Egg (d)	1.47 ± 0.01 (96) a	1.51 ± 0.04 (78) a	1.48 ± 0.01 (100) a	1.51 ± 0.01 (85) a	1.47 ± 0.01 (94) a	1.46 ± 0.01 (100) a	1.49 ± 0.01 (100) a
Larva (d)	0.99 ± 0.01 (85) cd	1.08 ± 0.03 (66) c	0.96 ± 0.01 (100) d	1.83 ± 0.04 (83) a	1.02 ± 0.01 (88) cd	1.25 ± 0.03 (90) b	1.21 ± 0.04 (75) b
Protonymph (d)	1.35 ± 0.03 (84) d	1.23 ± 0.04 (66) e	1.30 ± 0.03 (100) de	1.90 ± 0.04 (82) a	1.48 ± 0.01 (87) c	1.46 ± 0.04 (81) c	1.61 ± 0.04 (65) b
Deutonymph (d)	1.08 ± 0.02 (82) de	1.28 ± 0.03 (66) b	1.16 ± 0.02 (99) cd	1.02 ± 0.01 (82) e	1.44 ± 0.02 (87) a	1.18 ± 0.03 (76) c	1.10 ± 0.03 (62) cde
Pre-adult (d)	4.90 ± 0.03 (82) d	5.10 ± 0.01 (66) c	4.91 ± 0.03 (99) d	6.24 ± 0.05 (82) a	5.41 ± 0.02 (87) b	5.32 ± 0.05 (76) b	5.43 ± 0.06 (63) b
Pre-adult survival rate (*Sa*) ^§^	0.85 ± 0.04 (96) bc	0.85 ± 0.04 (78) bc	0.99 ± 0.01 (100) a	0.96 ± 0.02 (85) ab	0.93 ± 0.03 (94) bc	0.76 ± 0.04 (100) c	0.62 ± 0.05 (100) d
Female longevity (d)	48.20 ± 1.15 (41) a	50.36 ± 2.51 (33) a	39.26 ± 1.26 (50) b	40.46 ± 1.76 (41) b	20.38 ± 0.55 (45) d	27.72 ± 0.92 (43) c	29.90 ± 0.89 (42) c
Male longevity (d)	30.41 ± 0.77 (41) b	28.88 ± 1.57 (33) b	29.98 ± 1.31 (49) b	36.12 ± 1.68 (41) a	19.12 ± 0.52 (42) c	8.61 ± 0.25 (33) d	10.35 ± 0.42 (20) d
Total lifespan (d)	44.21 ± 1.22 (82) a	44.72 ± 1.96 (66) a	39.58 ± 1.02 (99) b	44.53 ± 1.24 (82) a	25.18 ± 0.39 (87) c	24.74 ± 1.21 (76) c	29.02 ± 1.31 (62) c
Total preoviposition period (TPOP) (d)	7.02 ± 0.14 (41) c	7.44 ± 0.19 (33) b	7.99 ± 0.25 (50) b	11.69 ± 0.37 (37) a	7.63 ± 0.10 (45) b	7.77 ± 0.16 (43) b	7.72 ± 0.17 (41) b
Adult preoviposition period (APOP) (d)	2.00 ± 0.13 (41) c	2.39 ± 0.11 (33) bc	3.06 ± 0.25 (50) b	5.32 ± 0.36 (37) a	2.18 ± 0.10 (45) c	2.44 ± 0.15 (43) bc	2.34 ± 0.15 (41) bc
Oviposition period (d)	25.80 ± 0.55 (41) a	19.61 ± 1.10 (33) b	18.14 ± 0.45 (50) b	7.54 ± 0.31 (37) d	12.57 ± 0.27 (45) c	7.67 ± 0.30 (43) d	6.83 ± 0.32 (41) d
Fecundity (eggs/female)	48.95 ± 0.97 (41) a	28.55 ± 2.37 (33) b	26.06 ± 0.70 (50) b	7.84 ± 0.34 (37) e	26.02 ± 0.69 (45) b	10.51 ± 0.46 (43) c	12.07 ± 0.45 (41) c

Values are mean ± SE (n). Data in the same row followed by different letters (a–e) are significantly different Tukey–Kramer HSD test (*p* < 0.05) results. ^§^ The standard errors were calculated using the bootstrap procedure with 100,000 bootstraps. The means followed by different letters (a–e) in the same row are significantly different between treatments using the paired bootstrap test (*p* < 0.05).

**Table 2 insects-13-01036-t002:** Life table parameters of *A. herbicolus* fed on six alternative foods and its natural prey, *T. urticae*.

Parameters	Natural Prey	Alternative Food					
	*T. urticae*	Apple Pollen	Almond Pollen	Maize Pollen	*O. bakeri*	*T. putrescentiae*	*A. ovatus*
*r* (day^−1^)	0.210 ± 0.011 a	0.155 ± 0.012 b	0.163 ± 0.008 b	0.064 ± 0.008 d	0.202 ± 0.010 a	0.120 ± 0.011 c	0.137 ± 0.012 bc
*λ* (day^−1^)	1.233 ± 0.014 a	1.168 ± 0.014 b	1.177 ± 0.010 b	1.066 ± 0.008 d	1.224 ± 0.012 a	1.128 ± 0.012 c	1.147 ± 0.014 bc
*R*_0_ (offspring/individual)	20.906 ± 2.494 a	12.077 ± 1.694 b	13.030 ± 1.348 b	3.412 ± 0.448 d	12.457 ± 1.384 b	4.520 ± 0.555 cd	4.950 ± 0.620 c
*T* (d)	14.509 ± 0.261 c	16.078 ± 0.558 b	15.708 ± 0.293 b	19.081 ± 0.620 a	12.488 ± 0.142 d	12.566 ± 0.339 d	11.693 ± 0.270 e

Values are mean ± SE. The standard errors were calculated using the bootstrap procedure with 100,000 bootstraps. The means followed by different letters (a–e) in the same row are significantly different between treatments using the paired bootstrap test (*p* < 0.05).

**Table 3 insects-13-01036-t003:** ScheirerRayHare statistical test (Cannibalism rate – Diet type × Predation days) results of cannibalism rate in R.

Cannibalism Rate	*df*	Sum sq	*H*	*p* Value
Diet type	6	18662687	455.80	0.00000
Predation days	6	350979	8.57	0.19912
Diet type: predation days	36	1034180	25.26	0.90970
Residuals	686	10005694		

## Data Availability

Data from this study are available from the first author upon reasonable request.

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
