# Peer review of "The Effects of Alternative Foods on Life History and Cannibalism of Amblyseius herbicolus (Acari: Phytoseiidae)"

_insects, 2022, doi:10.3390/insects13111036_

Round 1
Reviewer 1 Report
The authors conducted a set of simple experiments and showed that (among the 6 foods tested) Oulenziella bakeri and almond pollen are the best potential alternative foods for mass rearing of a predatory phytoseiid mite, Amblyseius herbicolus. The experiments were properly designed, conducted, and analyzed. The results of the study can be of some fundamental interest as an example of flexibility of food specificity in an omniphagous predator, but their applied significance is much higher, because these data can be used for the elaboration of optimal methods for mass rearing and field food supplementation of an important biocontrol agent. Thus, the manuscript can be published, although before publication it needs some improvements (see my comments below).
Lines 34-35: apple pollen ensured the highest fecundity (besides the natural prey T. urticae) and therefore it should be listed BEFORE (not AFTER) almond pollen and O. bakeri.
Line 59: consider replacing of “and can develop” with “that can develop” or “which can develop”.
Lines 78-79: I would not agree that “there were no suitable alternative foods for the mass rearing program”. This is too categorical and strong statement. In fact, a number of alternative foods were proposed and tested and many of them were suitable for A. herbicolus rearing (see earlier studies cited in this manuscript). I would rather conclude that “the known alternative foods were less suitable than the preferred natural prey” or something similar which is much closer to the reality.
Line 90: how long was A. herbicolus reared on O. bakeri before the experiment? If this rearing lasted for several or many generations, the predator can be selected for an ability to develop on this prey. Please, consider this issue here or in the Discussion.
Lines 195-196: This sentence is unclear. “P < 0.0001” means highly significant difference that is in contradiction with the previous “there was no significant difference between”. Possibly, P < 0.0001 is the result of the analysis of all dataset, not the listed 3 best foods? Please, correct, rewrite or explain.
Lines 201-205: Some of these data look a bit strange. Namely, feeding on maize pollen resulted in the longest protonymph duration and the longest total duration of pre-adult development but in the shortest deutonymph duration. How can you explain this? Please, consider this issue in the Discussion.
Line 206: replace “differences in eggs” with “differences in egg development duration” or something similar.
Line 207: replace “predatory eggs” with “predator eggs” or “eggs of the predator”.
Line 208: replace “lifespan of the predatory” with “lifespan of the predator” or “lifespan of the predatory mite”.
Line 217: replace “Development” with “Development duration”.
Line 218: delete “lowercase” because ONLY lowercase letters are used in this table.
Line 224 and Table 1: abbreviations APOP and TPOP are not explained. I do guess that APOP and TPOP are adult and total preoviposition periods, but this should be clearly explained in the text and in Table 1.
Figure 1: it is very difficult to see the data for the first days (pre-adult stages). I would suggest using of logarithmic time scale or, alternatively, data for pre-adult and adult stages can be presented in two figures with different time-scales.
Figure 3: the meaning of boxes, lines, and dots (e.g. means or medians, SD or quartiles, etc.) should be explained in the legend.
Lines 532-533: Please, use italics font for Latin names.
Author Response
Dear Reviewer,
Thank you very much for your helpful comments on our manuscript (ID: insects-2013836,Title: Effects of alternative foods on life history and cannibalism of Amblyseius herbicolus (Acari: Phytoseiidae)). We have modified our MS following your suggestion one by one, and the answers are as follows:
Response to Reviewer 1 Comments
Point 1: Lines 34-35, apple pollen ensured the highest fecundity (besides the natural prey T. urticae) and therefore it should be listed BEFORE (not AFTER) almond pollen and O. bakeri.
Response 1: We totally agree with the comment, and have revised our MS. Please see the manuscript (Lines 34).
Point 2: Line 59, consider replacing of “and can develop” with “that can develop” or “which can develop”.
Response 2: Thanks for your careful reminding, and we have changed the manuscript (Line 70).
Point 3: Lines 78-79, I would not agree that “there were no suitable alternative foods for the mass rearing program”. This is too categorical and strong statement. In fact, a number of alternative foods were proposed and tested and many of them were suitable for A. herbicolus rearing (see earlier studies cited in this manuscript). I would rather conclude that “the known alternative foods were less suitable than the preferred natural prey” or something similar which is much closer to the reality.
Response 3: Thank you for your advice, and we agree with the comment and have changed the manuscript (Line 91-92).
Point 4: Line 90, how long was A. herbicolus reared on O. bakeri before the experiment? If this rearing lasted for several or many generations, the predator can be selected for an ability to develop on this prey. Please, consider this issue here or in the Discussion.
Response 4: Thank you for your helpful suggestion. We have added the information of "2-3 generations" at Line 115.
Point 5: Lines 195-196, This sentence is unclear. “P < 0.0001” means highly significant difference that is in contradiction with the previous “there was no significant difference between”. Possibly, P < 0.0001 is the result of the analysis of all dataset, not the listed 3 best foods? Please, correct, rewrite or explain.
Response 5: Thanks for your helpful suggestion. We redescribed the results before the whole sentence to avoid misunderstanding (Line 244-245).
Point 6: Lines 201-205, Some of these data look a bit strange. Namely, feeding on maize pollen resulted in the longest protonymph duration and the longest total duration of pre-adult development but in the shortest deutonymph duration. How can you explain this? Please, consider this issue in the Discussion.
Response 6: Thank you for your reminder. It is a popular biological characteristic in predatory mites that the protonymph duration is much longer than deutonymph duration. It can be proved by: (1) the similar data of Neoseiulus californicus in the reference of Eini et al. (2022)(https://doi.org/10.1007/s10526-022-10129-7); (2) our own data: almost all data can show that the protonymph duration is much longer than deutonymph duration, although there is minor difference between different experimental groups.
Moreover, pre-adult duration is the sum of each undeveloped stages, and will be prolonged with protonymph duration becoming longer.
Point 7: Line 206, replace “differences in eggs” with “differences in egg development duration” or something similar.
Response7: Thanks for your suggestion. We have changed the manuscript (Line 258).
Point 8: Line 207, replace “predatory eggs” with “predator eggs” or “eggs of the predator”.
Response 8: Thank you. We have revised it. Please see the manuscript (Line 259).
Point 9: Line 208, replace “lifespan of the predatory” with “lifespan of the predator” or “lifespan of the predatory mite”.
Response 9: We have revised it. Please see the manuscript (Line 260-261).
Point 10: Line 217, replace “Development” with “Development duration”.
Response10: We have corrected it (Line 269).
Point 11: Line 218, delete “lowercase” because ONLY lowercase letters are used in this table.
Response 11: Thanks for your suggestion. We have changed the manuscript (Line 277).
Point 12: Line 224 and Table 1, abbreviations APOP and TPOP are not explained. I do guess that APOP and TPOP are adult and total preoviposition periods, but this should be clearly explained in the text and in Table 1.
Response 12: We have revised the MS according to your comments and added the full phrases of "TPOP" and "APOP " at Line 283-284 and Table 1.
Point 13: Figure 1, it is very difficult to see the data for the first days (pre-adult stages). I would suggest using of logarithmic time scale or, alternatively, data for pre-adult and adult stages can be presented in two figures with different time-scales.
Response 13: We appreciate your comment. We have drawn the survival rate curve of pre-adult and adult stages respectively to improve the image quality.
Point 14: Figure 3: the meaning of boxes, lines, and dots (e.g. means or medians, SD or quartiles, etc.) should be explained in the legend.
Response14: Thank you for your comments and we have revised it.
Point 15: Lines 532-533: Please, use italics font for Latin names.
Response 15: We have corrected it, please see the manuscript. (Line 635-636).
We hope the changes to meet your requirements for publication.
Thank you for your help.
Yours sincerely,
All authors,
Institute of Entomology
Guizhou University

Reviewer 2 Report
Review report on submission Insects-2013836
The manuscript presents results of laboratory experiments which tested the effects of several types of alternative food including plant pollen on the life history and population growth parameters of phytoseiid mite Amblyseius herbicolus and compares them to natural prey, two spotted spider mite. Although many studies on the role of alternative food for Phytoseiidae have been published, the present study deals with a new perspective biocontrol agent A. herbicolus (search for topic on this species in Web of Science returns only 37 records compared to e.g. 363 records on A. swirskii). The effect of food quality on cannibalism in A. herbicolus has been studied recently by Marcossi et al. (2020) but that study used pollen only. The study reported in the evaluated manuscript is much wider as more alternative foods were used and besides cannibalism, the developmental, reproduction and life table parameters were evaluated, too. The manuscript is well written and fits well into the scope of Insects. Methods used are well described with all important details, numbers of replications are sufficient, data analysed by appropriate statistics and results presented and discussed well. I have only few suggestions for minor revision (see below, numbers indicate lines).
46 I usually recommend to write key words other than those written in title, e.g. in this study relevant could be: “biological control”, “mass rearing”, “pollen”, “food quality”
74-76 It will be good to include authors of species names and specify higher taxons (Order: Family)
79-80 Please write genera name in full when mentioned for the first time in ms besides Abstract and add author names and higher taxons
81 – the same as above
89 symbol for degree seems to be incorrect, please add space before E)
91 25°C ± 2°C, 85% ± 5% > rewrite as 25±2 °C, 85±5%
97 – as suggested above
100 should read “at –20°C in a freezer”
103 rewrite as 60±10% RH
103 Are you sure about 12:12 h light/dark photoperiod? Under such short day conditions T. urticae normally enters diapause so no rearing is possible. It seems to be a mistake, please check!
142 correct to 25±2 °C, 85±5% RH
150 should read: The number of cannibalized eggs was recorded daily
151 Please specify microscope used (brand, model, manufacturer city and country, e.g. (Olympus SZ61, Tokyo, Japan) and magnification
155 – see comments above
158 “that” should not be italics
195-196 there is some mistake, when P<0.0001 than it is statistically significant, I gues it is the effect of food type while post-test revealed no differences among O. bakeri, T. putrescentiae and A. ovatus. This needs to be clarified/rewritten.
Table 1 column 1: Oviposition days (d) will be better written as Oviposition period (d)
245 while on its natural prey
310 Please write genera name in full and add author names as these species are mentioned for the first time in ms.
312 add name of author who described the species
334 offered a list of plant pollen species
355 delete years from citations
Author Response
Dear Reviewer,
Thank you very much for your helpful comments on our manuscript (ID: insects-2013836,Title: Effects of alternative foods on life history and cannibalism of Amblyseius herbicolus (Acari: Phytoseiidae)). We have modified our MS following your suggestion one by one, and the answers are as follows:
Response to Reviewer 2 Comments
Point 1: 46, I usually recommend to write key words other than those written in title, e.g. in this study relevant could be: “biological control”, “mass rearing”, “pollen”, “food quality”
Response 1: Thanks for your careful reminding and have changed the manuscript (Line 46).
Point 2: 74-76, It will be good to include authors of species names and specify higher taxons (Order: Family)
Response 2: Thanks for your careful reminding and have modified the manuscript (Line 83-89).
Point 3: 79-80, Please write genera name in full when mentioned for the first time in ms besides Abstract and add author names and higher taxons
Response 3: Thanks for your careful reminding and have changed the manuscript (Line 93-96).
Point 4: 81, – the same as above
Response 4: We have modified it in the manuscript (Line 96).
Point 5: 89, symbol for degree seems to be incorrect, please add space before E)
Response 5: Thank you for your comments and we have revised it. Please see the manuscript (Line 114).
Point 6: 91, 25°C ± 2°C, 85% ± 5% > rewrite as 25±2 °C, 85±5%
Response 6: We have revised it, please see the manuscript. (Line 122).
Point 7: 97, – as suggested above
Response 7: We have revised it, please see the manuscript. (Line 122)
Point 8: 100, should read “at –20°C in a freezer”
Response 8: Thanks for your careful reminding and have changed the manuscript (Line 125).
Point 9: 103, rewrite as 60±10% RH
Response 9: Thank you for your suggestion and we have revised it. Please see the manuscript (Line 128).
Point 10: 103, Are you sure about 12:12 h light/dark photoperiod? Under such short day conditions T. urticae normally enters diapause so no rearing is possible. It seems to be a mistake, please check!
Response 10: Thanks for your careful reading, it is realy a mistake has been corrected as “14:10 h light/dark photoperiod” (Line 128).
Point 11: 142, correct to 25±2 °C, 85±5% RH
Response 11: We have revised it, please see the manuscript (Line 182).
Point 12: 150, should read: The number of cannibalized eggs was recorded daily
Response 12: Thanks for your careful reminding. We have changed the manuscript (Line 190).
Point 13: 151, Please specify microscope used (brand, model, manufacturer city and country, e.g. (Olympus SZ61, Tokyo, Japan) and magnification
Response 13: we have modified the MS following your suggestion (Line 191).
Point 14: 155, – see comments above
Response 14: Thank you for your comments and we have revised it. Please see the manuscript (Line 195).
Point 15: 158, “that” should not be italics
Response 15: We have revised it, please see the manuscript (Line 199).
Point 16: 195-196, there is some mistake, when P<0.0001 than it is statistically significant, I guess it is the effect of food type while post-test revealed no differences among O. bakeri, T. putrescentiae and A. ovatus. This needs to be clarified/rewritten.
Response 16: Thanks for your helpful suggestion. We now presented the results first before the whole sentence to avoid misunderstanding (Line 244-245).
Point 17: Table 1 column 1: Oviposition days (d) will be better written as Oviposition period (d)
Response 17: Table 1 column 1 has been modified. Please see the manuscript.
Point 18: 245, while on its natural prey
Response 18: Thanks. We have revised the manuscript (Line 305).
Point 19: 310, Please write genera name in full and add author names as these species are mentioned for the first time in ms.
Response 19: Thanks for your suggestion. We have modified it (Line 386-387).
Point 20: 312, add name of author who described the species
Response 20: Thanks for your careful reminding. We have changed the manuscript (Line 389).
Point 21: 334, offered a list of plant pollen species
Response 21: Thanks. We have corrected it (Line 411).
Point 22: 355, delete years from citations
Response 22: Thank you for your suggestion and we have revised it. Please see the manuscript (Line 437-438).
We hope the changes to meet your requirements for publication.
Thank you for your help.
Yours sincerely,
All authors,
Institute of Entomology
Guizhou University